

# RBM25 binds to and regulates alternative splicing levels of *Slc38a9, Csf1,* and *Coro6* to affect immune and inflammatory processes in H9c2 cells

Xin Tian[1], Guangli Zhou[1], Hao Li[1], Xueting Zhang[1], Lingmin Zhao[1], Keyi Zhang[1], Luqiao Wang[1], Mingwei Liu[1], Chen Liu[2] and Ping Yang[1]

[1] Department of Cardiology, The First Affiliated Hospital of Kunming Medical University, Kunming, China
[2] Department of Radiology, Affiliated Hospital of Yunnan University, Kunming, China

## ABSTRACT

**Background**. Alternative splicing (AS) is a biological process that allows genes to be translated into diverse proteins. However, aberrant AS can predispose cells to aberrations in biological mechanisms. RNA binding proteins (RBPs), closely affiliated with AS, have gained increased attention in recent years. Among these RBPs, RBM25 has been reported to participate in the cardiac pathological mechanism through regulating AS; however, the involvement of RBM25 as a splicing factor in heart failure remains unclarified.

**Methods**. RBM25 was overexpressed in H9c2 cells to explore the target genes bound and regulated by RBM25 during heart failure. RNA sequencing (RNA-seq) was used to scrutinize the comprehensive transcriptional level before identifying AS events influenced by RBM25. Further, improved RNA immunoprecipitation sequencing (iRIP-seq) was employed to pinpoint RBM25-binding sites, and RT-qPCR was used to validate specific genes modulated by RBM25.

**Results**. RBM25 was found to upregulate the expression of genes pertinent to the inflammatory response and viral processes, as well as to mediate the AS of genes associated with cellular apoptosis and inflammation. Overlap analysis between RNA-seq and iRIP-seq suggested that RBM25 bound to and manipulated the AS of genes associated with inflammation in H9c2 cells. Moreover, qRT-PCR confirmed *Slc38a9*, *Csf1*, and *Coro6* as the binding and AS regulatory targets of RBM25.

**Conclusion**. Our research implies that RBM25 plays a contributory role in cardiac inflammatory responses via its ability to bind to and regulate the AS of related genes. This study offers preliminary evidence of the influence of RBM25 on inflammation in H9c2 cells.

## INTRODUCTION

Heart failure (HF) is a burgeoning global epidemic with a 5-year mortality rate of approximately 50%. HF affects 13.7 million individuals in China and more than 37.7 million people globally, with increasing incidence (*McMurray & Pfeffer, 2005*; *Hao et al., 2019*;

Corresponding authors
Chen Liu, 25997055@qq.com
Ping Yang, 15877990331@163.com

*Groenewegen et al., 2020*). Salient risk factors that foster an augmented susceptibility to HF include hypertension, diabetes mellitus, metabolic syndrome, and atherosclerotic disease (*Yancy et al., 2013*). In the setting of HF, the left ventricle (LV) exhibits an immoderate plasticity response to multiple insults, including myocardial anoxia/ischemia and increased wall stress; this process is known as pathological LV remodeling, which serves as a principal catalyst for both HF and its poor prognosis (*Hill & Olson, 2008*). Cellular-level events implicated in LV remodeling include cardiac myocyte apoptosis, hypertrophic growth, hyperplasia, fibrosis, and inflammation (*Burchfield, Xie & Hill, 2013*). Therefore, the cornerstone for improvement of HF therapy and prognosis rests on a robust understanding of the core mechanisms involved.

An increasing body of evidence highlights the pivotal role of systemic inflammation in HF pathophysiology. Inflammation is implicated not only in HF development and progression but also in its complications (*Murphy et al., 2020*). Additionally, inflammation can predict poor prognosis beyond traditional indices such as left ventricular ejection fraction and New York Heart Association functional class (*Schiattarella, Rodolico & Hill, 2021*).

Based on this, several strategies aimed at mitigating pro-inflammatory cytokines in HF have been proposed. However, most studies have yielded conflicting results, with most proving unsuccessful (*Mann, 2015*). Only a limited number of tests, notably intravenous immunoglobulin therapy targeting the neutralization of autoantibody titers, have shown promise (*Gullestad et al., 2001*; *McNamara et al., 2001*). This knowledge gap underscores our confined understanding of the inflammatory pathways in HF and the potential influence of inflammation on HF therapy. Thus, delineating the pathways through which inflammation either exacerbates or alleviates HF will serve to enhance our ability to reduce the incidence and severity of this chronic condition.

RNA-binding proteins (RBPs) bind to various forms of RNA in targeted genes, such as pre-mRNA, mRNA, and non-coding RNA, which enables them to regulate mRNA or protein expression levels across a spectrum of cells (*Hentze et al., 2018*). These RBPs affect different aspects of RNA, including splicing, transport, translation, and decay (*Tiedje et al., 2016*). RBPs serve as regulators in an assortment of biological processes, from germline and early embryo development to immune and inflammation reactions in cancer and other diseases (*Fu & Blackshear, 2017*; *Turner & Díaz-Muñoz, 2018*). Interest around RBPs has spiked as more of them, including RBM24 and QKI, have been found to play roles in the formation, morphogenesis, and maturation of the heart (*Maatz et al., 2014*; *Chen et al., 2021*). Additionally, several cardiovascular ailments and dysfunctions in cardiomyocytes, ranging from defects in fetal heart development to denuding of the coronary artery endothelium, are linked to RBPs (*Blech-Hermoni & Ladd, 2013*; *De Bruin et al., 2017*).

RBM25 is an RBP that shares the RE/RD-rich (ER) central region and C-terminal proline-tryptophan-isoleucine motif (*Fortes et al., 2007*). Previous studies have linked human HF to sodium voltage-gated channel alpha subunit 5 (SCN5A) terminal exon splicing variants. The pre-mRNA splicing gene *LUC7L3* is a homolog of the yeast U1 snRNP-associated factor in humans. In HF, it has been demonstrated that the increase in *LUC7L3* and RBM25 can improve RBM25-SCN5A mRNA binding, SCN5A splice variant

abundances and finally decrease the $Na^+$ current (*Zhou et al., 2011*). Despite serving as a pre-mRNA processor regulating the pre-mRNA of SCNA5, a protein pivotal in HF development, the regulatory mechanism of RBM25 in HF is still unclear.

In the present study, we hypothesized that overexpression of RBM25 could induce changes in the transcription level and AS events by binding to relevant genes in H9c2 cells. Through harnessing RNA-seq and iRIP technology, we explored target genes that are potentially bound and regulated by RBM25, with the aim to gain insights into the molecular mechanism of RBM25 in H9c2 cells.

## MATERIALS AND METHODS

This project was approved by the Institutional Ethics Committee at the First Affiliated Hospital of Kunming Medical University (kmmu20211238, Yunnan, China).

### Cloning and plasmid construction

pCDNA3.1-3xflag-RBM25 was provided by Youboit Biotech (Changsha, China).

### Cell culture and transfection

The H9c2 cell line (CL-0089), purchased from Procell Life Science & Technology Co., Ltd. Wuhan, China, was grown in DMEM at 37 °C with 5% $CO_2$, 10% fetal bovine serum (FBS), 100 U/mL 1% streptomycin and penicillin. The plasmid was transfected into H9c2 cells using Lipofectamine 2000 (Invitrogen, Carlsbad, CA, USA), according to the manufacturer's instructions. Briefly, 20 µL of LipofectamineTM 2000 transfection reagent (11668019, Invitrogen, Shanghai, China) and 1,000 ng of plasmid were added to 20 µL of serum-free medium and cultured for 5 min. Subsequently, the plasmid was added to the transfection solution and allowed to grow for 20 min. Next, the mixture was evenly poured into DMEM and cultured for 6 h at 37 °C with 5% $CO_2$. For RT-qPCR and western blotting analysis, the transfected cells were taken 48 h after the medium had been changed (Nasci VL 2019).

### RNA extraction and quantitative real-time PCR (qRT-PCR)

To evaluate the effects of RBM25 overexpression, we used glyceraldehyde-3-phosphate dehydrogenase (GAPDH) as a control gene. Standard cDNA synthesis protocols were followed, and RT-qPCR using the HieffTM qPCR SYBR® Green Master Mix (11202ES08, Low Rox Plus; YEASEN, Shanghai, China) was conducted on a Bio-Rad S1000. The primers used in the study are listed in Table 1. The level of GAPDH mRNA was used to normalize the concentration of each transcript. The paired Student's $t$-test was used to compare data using the GraphPad Prism program (San Diego, CA, USA).

### Western blotting

Western blotting was conducted according to previously described protocols (*Huang et al., 2021*). H9c2 cells were lysed in an ice-cold wash buffer (phosphate buffered saline (PBS) with 0.1% SDS, 0.5% NP-40, and 0.5% sodium deoxycholate) supplemented with a protease inhibitor cocktail (Roche, Basel, Switzerland) and incubated for 30 min on ice. After boiling the samples for 10 min with 1X SDS sample buffer, we separated them on a

**Table 1  The primer sequences used for validation by real-time qPCR.**

| | |
|---|---|
| Rat-GAPDH-F | TCTCTTGTGACAAAGTGGACA |
| Rat-GAPDH-R | CCCATTCTCAGCCTTGACTGT |
| Cd28-F | ATGTCTGTCTGGTGTCTCA |
| Cd28-R | GGTTAATGGTGGCTATATGGA |
| Cxcl10-F | AGTTCTGAGTTCACCTGAG |
| Cxcl10-R | CCTGTGAGTATTCTGAGTTATG |
| Dhx58-F | CTGGCTTGCTGAACTCTC |
| Dhx58-R | TACCTGACTTCGACATTCTG |
| Ifi47-F | AGGATGACAGCATAATAGCA |
| Ifi47-R | CGACCACCACATTAAGGAA |
| Ifit1-F | CGTTTGGCTTACTGGAGAA |
| Ifit1-R | ACATAGAGGTGCGTCCTT |
| Ifit3-F | GCTGGATTGTTGGCATTTC |
| Ifit3-R | CGGTTGTTATCAGGCTCAT |
| Irf7-F | GACATATAGCCAAGGAATAAGC |
| Irf7-R | CTCTACGATGACATTGAACAC |
| Irgm2-F | ATATGTTGTGGGTAAGGATAGG |
| Irgm2-R | AAGAACGAGCAGGCAGAA |
| Lnc215-F | CTTCAATGAGACAGTTAAGC |
| Lnc215-R | GATCCATGAGCAGGACTAT |
| Mx2-F | CAGGAATCAGGTGACTTCA |
| Mx2-R | CCTTACTTCTGGTCCAATTAC |
| Oas1a -F | TAGAGTGAAGTTTGAGGTCC |
| Oas1a -R | CATAGGCTGGAAGCACAT |
| Oas1b -F | CACACCTCCTCCAACAAG |
| Oas1b -R | CCTTCCTTACAACCTGCTT |
| Slc38a9-M-F | AGCAGCCCTGTGATCTGTCC |
| Slc38a9-AS-F | ACGTGATTGATGATCTGTCC |
| Slc38a9-M/AS-R | TTGAGCAGCGGCAAGAGGAG |
| Csf1-M-F | CTAATGAATGCTCGTTCTGT |
| Csf1-AS-F | GGACACAGGCCTCGTTCTGT |
| Csf1-M/AS-R | GACGATCCCGTTTGCTACCT |
| Il17rc –M/AS-F | GGTACGAATCTGGTCCTACAC |
| Il17rc -AS-R | GAGGTCCAGTCAGGCAACTG |
| Il17rc -M -R | CCCCTGCAGTCAGGCAACTG |
| Coro6-M/AS-F | TGCGAAACATCACAGAACCT |
| Coro6-AS-R | CCAACACACGCTGCACTGAG |
| Coro6-M-R | TTATCACCACCTGCACTGAG |
| Brd8-M/AS-F | CACTGGTTGCTGGCTGTATGTC |
| Brd8-AS-R | TACCTCTGGAGTGAATGAGA |
| Brd8-M-R | CCCCAGACTGGTGAATGAGA |

10% SDS-PAGE gel. PVDF membranes were blocked for 1 h at room temperature, before incubating overnight at 4 °C with primary antibodies against FLAG (anti-FLAG, 1:1000, antibody produced in rabbit, 2368s, CST, San Antonio, TX, USA) and GAPDH (1:5000, antibody produced in mouse, 60004-1-Ig, Proteintech, China). Following incubation, the membranes were incubated with horseradish peroxidase-conjugated secondary antibodies (anti-rabbit, 1:10000, SA00001-2, Proteintech, China and anti-mouse, 1:10,000, AS003, Abclonal, Beijing, China) for 45 min at room temperature. Finally, the membranes were visualized using enhanced ECL reagent (P0018FM, Beyotime, Shanghai, China) through chemiluminescence.

## High-throughput sequencing

The RNA-seq libraries for each sample were prepared using the KAPA Stranded mRNA-Seq Kit for Illumina® Platforms (KK8544, San Diego, CA, USA), which required 1 g of total RNA. VAHTS mRNA capture beads (N401-01) were used for polyadenylated mRNA isolation. The isolated mRNAs were fragmented using iron at 95 °C. Subsequently, their ends were repaired and 5′ adaptors were ligated. Double-stranded cDNA was synthesized from the fragmented mRNAs, which were then quantified, amplified, and stored at −80 °C until sequencing (*Li et al., 2019*). High-throughput sequencing was performed using the Illumina Novaseq 6000 equipment for 150-nt paired-end sequencing following the manufacturer's instructions (Ablife Inc., Wuhan, China). The average sequencing base was approximately 10G per sample, resulting in a sequencing depth of 100 (10G/100M). Three biological replicates were conducted, and the biological coefficient of variation in the group was calculated as 0.0392 based on edgeR software. The target effect size was set at 2, and the statistical power of this experimental design was 1.00 according to the calculation conducted in RNASeqPower.

## Data cleaning and alignment

We removed any reads containing more than 2-N bases from the raw sequencing reads. Adaptors and low-quality bases were trimmed using the FASTX-Toolkit (Version 0.0.13), and reads shorter than 16 nt were filtered. The mRatBN7.2 genome was subsequently aligned by employing cleaned reads and HISAT2 with a tolerance of four mismatches (*Kim, Langmead & Salzberg, 2015*). To determine the expression level of each gene, the fragments per kilobase of transcript per million fragments mapped (FPKM) metric was calculated based on uniquely mapped reads.

## Analysis of differentially expressed genes (DEGs)

Potential DEGs were filtered using the R Bioconductor tool DESeq2, applying the criteria of fold change ≥ 2 or ≤ 0.5 and a false discovery rate (FDR) < 0.05 (*Robinson, McCarthy & Smyth, 2010*).

## Alternative splicing analysis

The ABLas pipeline (*Kim et al., 2013*) was employed to ascertain alternative splicing events (ASEs) and regulated alternative splicing events (RASEs). Briefly, ASEs, including exon skipping, cassette exons, alternative 3′ and 5′ splice sites, mutual exon skipping (MXE),

MXE paired with an alternative polyadenylation site (3pMXE), and MXE combined with an alternative 5′ promoter (5pMXE), were identified based on splice junction reads. The significance level for Student's $t$-test was set at $p < 0.05$. ASE ratios in RNA-seq were calculated as alternative junction reads / (alternative junction reads + model junction reads), while the ASE ratios in qRT–PCR were determined as the alternative splicing transcript level/model transcript level.

### Improved RNA immunoprecipitation sequencing (iRIP-seq)

For immunoprecipitation, 15 μL of RBM25-antibody (25297-1-AP, Proteintech, Chicago, IL, USA) and control IgG-antibody (AC005, ABClonal, Wuhan, China) were added to cell lysis buffer and incubated overnight at 4 °C. Subsequently, Protein A/G Dynabeads (Thermo Scientific, Waltham, MA, USA) were added to the immunoprecipitate and incubated for 2 h at 4 °C. Following application to a magnet and removal of the supernatant, the beads were successively washed twice with lysis buffer, once with PNK buffer, and once with high-salt buffer (250 mM Tris, 750 mM NaCl, 10 mM EDTA, 0.1% SDS, 0.5% NP-40, and 0.5 deoxycholates, pH = 7.4).

Subsequently, the beads were resuspended in elution buffer (50 nM Tris, 10 mM EDTA, and 1% SDS, pH = 7.4) and incubated for 20 min at 70 °C in a heat block, before removing the magnetic beads from the separator. Next, the supernatant was transferred to a clean, 1.5-mL microfuge tube. At a final concentration of 1.2 mg/mL, proteinase K (Roche, Basel, Switzerland) was added to the 1% input (without immunoprecipitation) and RBP was immunoprecipitated with crosslinked RNA. Then, a 120-min cultivation was performed at 55 °C. TRIzol reagent (15596-018, Ambion, Austin, TX, USA) was used to purify the RNA.

### Real-time qPCR validation of DEGs, ASEs, and iRIP

To quantitatively analyze the two distinct splicing isoforms of a particular ASE using the qPCR technique, two pairs of primers that matched the splice junctions of the constitutive and alternative exons were designed to selectively amplify each isoform. GAPDH was used as a control gene. To quantify the relative expression level of genes selected from the iRIP analytic outcomes, the primer-binding site was employed as a control gene to perform real-time qPCR. Standard cDNA synthesis techniques were followed, and RT-qPCR was conducted on a Bio-Rad S1000 using Hieff qPCR SYBR® Green Master Mix (11202ES08, Low Rox Plus; YEASEN, Shanghai, China). The primer information is shown in Table 1. The amount of mRNA was determined using the $2^{-\Delta\Delta CT}$ method.

### Functional enrichment analysis

Employing the Kyoto Encyclopedia of Genes and Genomes (KEGG) Orthology Based Annotation System (KOBAS) 3.0 (*Xie et al., 2011*), a web server for gene and protein functional annotation and functional enrichment, we next sought to investigate the functional categories through the identification of Gene Ontology (GO) keywords and KEGG pathways. Benjamini–Hochberg FDR adjustment and the hypergeometric test were used in the analytical procedure. To identify significantly enriched terms, we implemented an adjusted $p$-value cutoff of <0.05.

## Statistical analyses

All data are presented as the mean ± standard deviation (SD) and were processed by SPSS 16.0 statistical software (Chicago, IL, USA) and GraphPad Prism 8.0 (San Diego, CA, USA). The Student's $t$-test was used for a two-group comparison, where a $p$-value $<0.05$, two-tailed, was deemed statistically significant. Experiments were performed three times independently.

## RESULTS

### RBM25 is successfully overexpressed in H9c2 cells

Cells transfected with RBM25 were subjected to analysis to identify overexpression using RT-qPCR and western blotting techniques. As depicted in Fig. 1, the expression levels of RBM25 mRNA and protein were increased in the overexpression (OE) group compared to the control group.

### RBM25 upregulates the expression of inflammatory response and viral response associated genes in H9c2 cells

To investigate the molecular mechanisms underlying RBM25 functions in H9c2 cells, we employed RNA-seq to analyze the gene expression profiles of RBM25-OE cells and control cells, with two biological replicates for each condition. The mean Q30 of RBM25-OE and the control group was 94.81% and 94.84%, respectively. After removal, an average of 70.43 million clean pair-end reads per sample were obtained, which were then mapped to the rat genome mRatBN 7.2. This resulted in an average of 62.2 million uniquely mapped reads, which were used to calculate gene expression levels measured by FPKM. As a result, 16,749 genes were expressed with FPKM >0, among which, 10,195 demonstrated an expression level of FPKM >1 in at least one sample.

Principal component analysis based on FPKM values illustrated that the control and RBM25-OE samples could be differentiated (Fig. 2A). Using fold change ≥ 1.5 or ≤ 0.67 and a 5% FDR as the cut-off value for distinguishing significant DEGs, 80 significant DEGs relating to RBM25 overexpression were identified, among which, 49 genes were upregulated and 31 were downregulated (Fig. 2B). A heatmap of the hierarchical clustering of DEG expression levels identified a high similarity within replicated groups and clearly differentiated between RBM25-OE samples and control samples (Fig. 2C).

We next implemented GO function analysis to further explore the potential biological roles of these 80 DEGs. We found that the upregulated DEGs in the RBM25-OE cells were significantly enriched in viral response and metabolic processes (Fig. 2D). The top ten outcomes of the KEGG pathway analysis are illustrated in Fig. S1. For downregulated genes, the main KEGG pathways were *Staphylococcus aureus* infection, type I diabetes mellitus, complement and coagulation cascades, and nitrogen metabolism. However, no GO terms were enriched. The hierarchical clustering heatmap shown in Fig. 2E presents the FPKM of the inflammatory response- and viral response-associated genes.

The expression of DEGs known to be implicated in HF, such as *Lnc215*, *Irf7*, *Cxcl10*, *Oas1b*, *Irgm2*, *Mx2*, *Ifi47*, *Ifit1*, *Oas1a*, *Ifit3*, *Cd28*, and *Dhx58* (full names provided in Table S1), were further validated through RT-qPCR. As shown in Fig. 2F, all selected genes
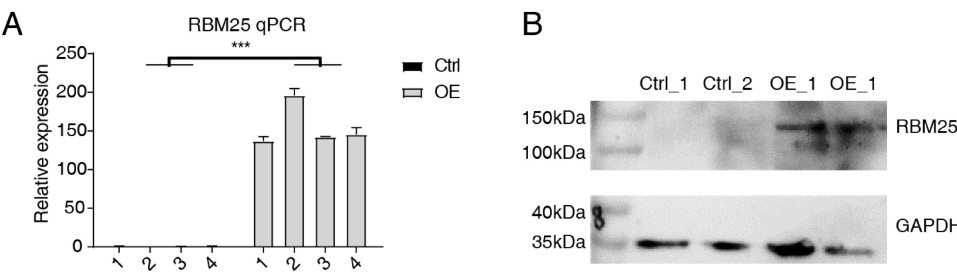

**Figure 1** **RBM25 overexpression in H9c2 cells.** (A) RT-qPCR results of RBM25-OE and the control samples. (B) Western blot of RBM25 overexpression. *P* values were determined by paired Students' t test. \**P* value < 0.05, \*\**P* value < 0.01, \*\*\**P* value < 0.001.

displayed a significant increase in the RBM25-OE group compared to the control group, which aligns with the results of RNA-seq analysis.

## RBM25 regulates alternative splicing of cellular inflammation- and apoptosis-associated genes in H9c2 cells

Considering the major biological role of RBM25 in regulating AS, we next investigated whether RBM25 affected DEGs by modulating the AS process. After analyzing the splice junction using HISAT2 in our sequenced transcriptome data, we obtained 301,474 junctions; approximately 51% (153,758 of 301,474) of these were known splice junctions, while nearly 49% (147,716 of 301,474) were novel splice junctions. We then used the ABlas software to evaluate the ASEs for these splice junctions, which produced a total of 37,817 ASEs, among which, 49.08% were novel. The numbers of all significant ASEs are summarized in Fig. S2. High-confidence RBM25 RASEs were further identified by selecting an ASE ratio alteration reaching statistical significance ($p \leq 0.05$). Following this criterion, 700 high-confidence RASEs were identified. The numbers of each AS type are summarized in Fig. 3A. Functional enrichment analysis was then performed, and analysis of GO terms revealed that RBM25-regulated alternative splicing genes (RASGs) were highly enriched in processes such as protein phosphorylation, apoptotic process, and positive regulation of transcription, as shown in Fig. 3B. KEGG pathway enrichment analysis revealed primary involvement in processes such as ubiquitin-mediated proteolysis, mitophagy, N-Glycan biosynthesis, phosphonate and phosphinate metabolism, neurotrophin signaling pathway, mitogen-activated protein kinase (MAPK) signaling pathway, salmonella infection, RNA degradation, Wnt signaling pathway, and regulation of actin cytoskeleton. However, an analysis of the overlap between the DEGs from the RNA-seq data and the RASGs revealed no common genes, suggesting that RBM25 does not simultaneously influence both the transcript levels and AS of most DEGs in H9c2 cells. Further analysis of the AS pattern between the control and RBM25-OE cells *via* RT-PCR showed that the alternative 5′ splice site and exon skipping were representative of the affected ASEs, as illustrated in Figs. 3C–3D.

Among all these ASEs, we selected five RASGs known to be involved in HF, namely *Slc38a9*, *Csf1*, *Il17rc*, *Coro6*, and *Brd8* (*Shaposhnik, Wang & Lusis, 2010*; *Wyant et al., 2017*;

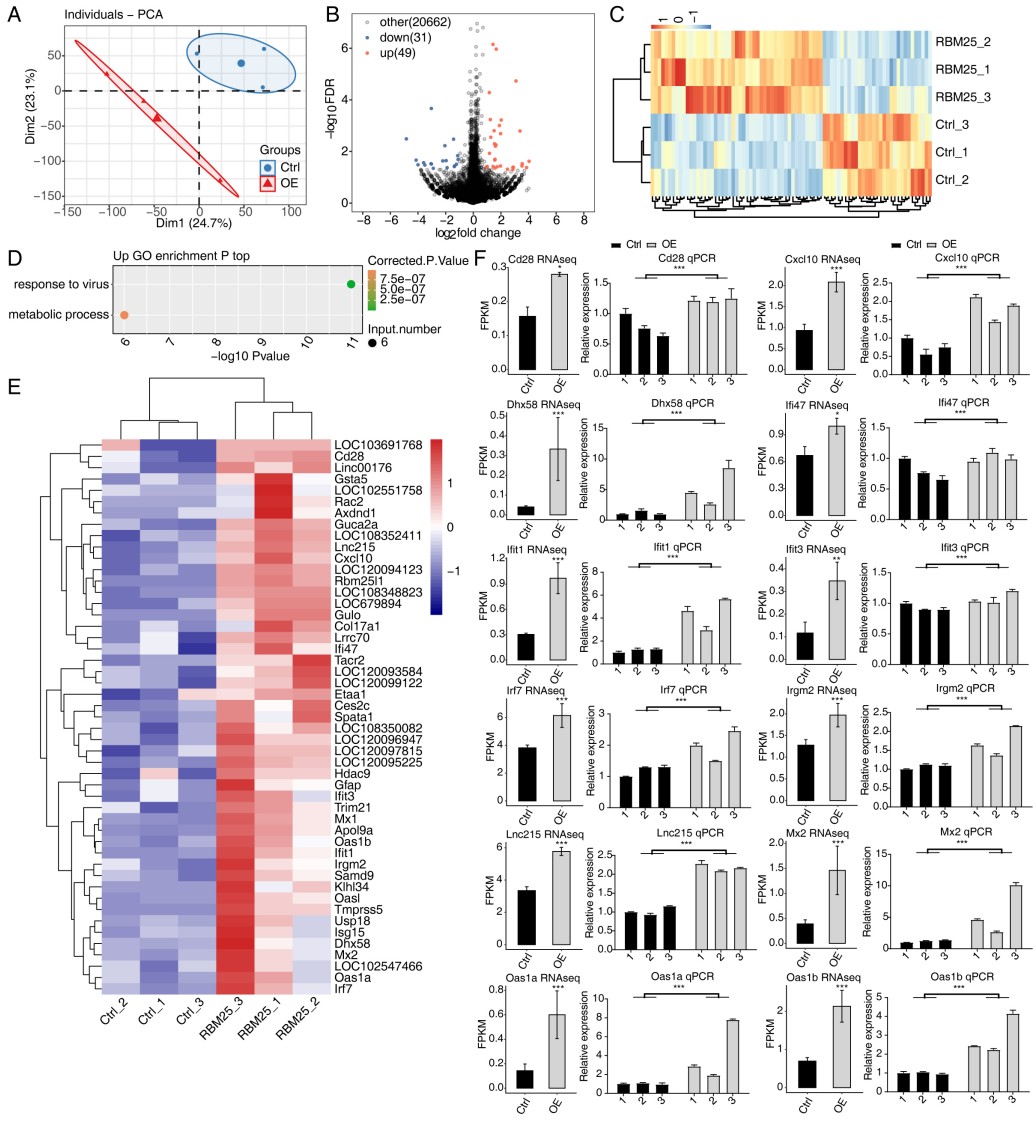

**Figure 2 RBM25 regulates gene expression in H9c2 cells.** (A) Principal component analysis (PCA) based on the FPKM value of all detected genes. The ellipse for each group is the confidence ellipse. (B) Volcano plot showing all differentially expressed genes (DEGs) between the RBM25-OE and the control samples with DESeq2. *P* value < 0.05 and FC (fold change) ≥ 1.5 or ≤ 0.67. (C) Hierarchical clustering heat map showing expression levels of all DEGs. (D) Bubble diagram exhibiting the most enriched GO biological process results of the up-regulated DEGs. (E) Hierarchical clustering heat map showing expression levels of inflammatory response and virus response associated genes. (F) Bar plot showing the expression pattern and qPCR validation results of selected genes. Error bars represent mean ± SEM. *P* value < 0.05, **P* value < 0.01, ***P* value < 0.001.

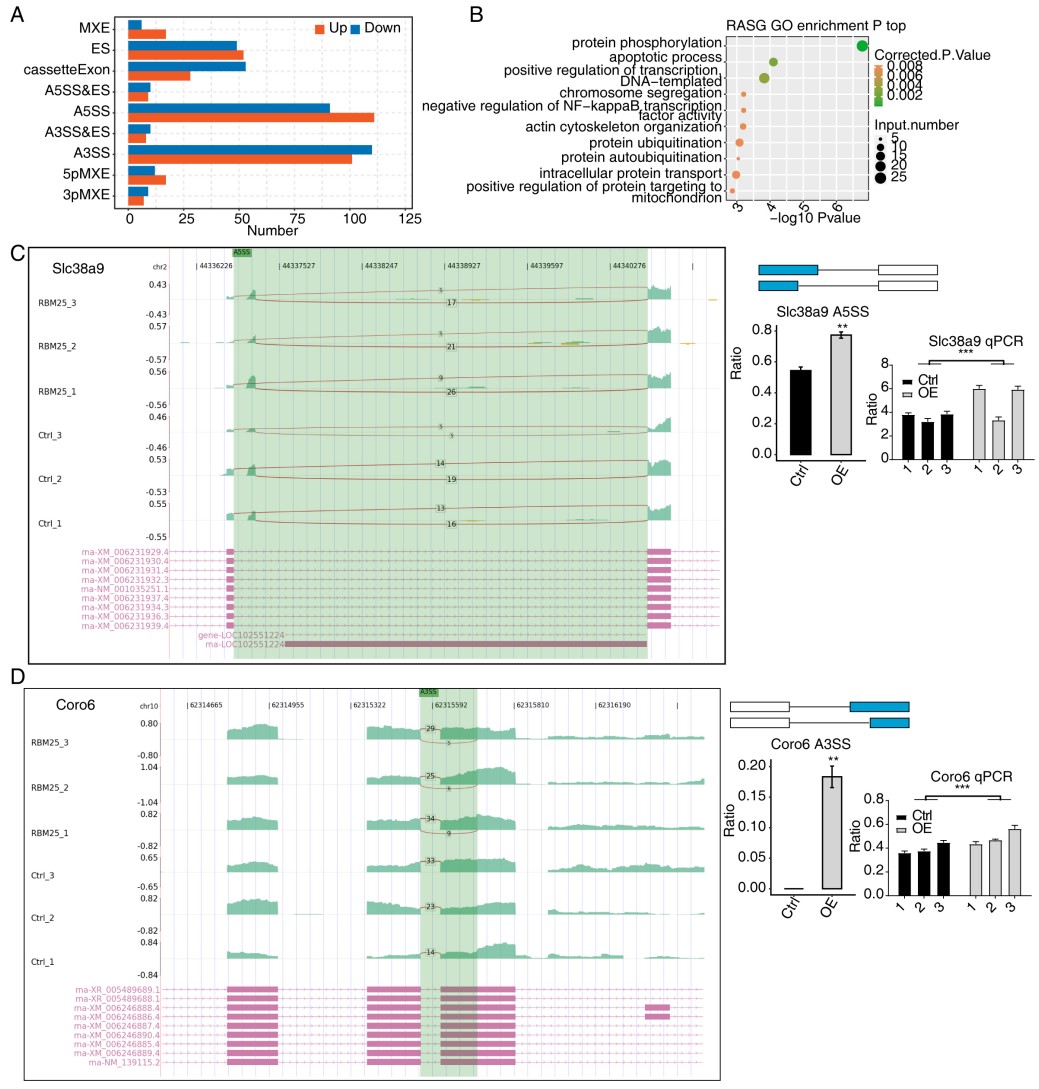

**Figure 3  RBM25 regulates gene alternative splicing in H9c2 cells.** (A) Bar plot showing the number of all significant regulated alternative splicing events (RASEs). Up and down represented the higher and lower alternative splicing (AS) ratios in RBM25-OE samples than in the control sample, respectively. (B) Bubble diagram exhibiting the most enriched GO biological process results of the regulated alternative splicing genes (RASGs). (C–D) RBM25 regulates alternative splicing of *Slc38a9* (C), and *Coro6* (D). Left panel: IGV-sashimi plot showing the regulated alternative splicing events and binding sites across mRNA. Reads distribution of RASE is plotted in the up panel, and each gene's transcripts are shown below. Right panel: The schematic diagrams depict the structures of ASEs. RNA-seq and qPCR validation of ASEs are shown at the bottom of the right panel. Error bars represent mean ± SEM. *P value < 0.05, **P value < 0.01, ***P value < 0.001.

*Chang et al., 2018*; *Sciarretta et al., 2018*; *Van den Hoogenhof et al., 2018*; *Ren et al., 2022*), with the aim to validate the effects of RBM25 on alternative splicing. It appeared that the levels of *Slc38a9, Csf1*, and *Coro6* significantly increased following RBM25 overexpression, while the level of *Brd8* decreased. In contrast, there was no statistically significant difference in the level of *Il17rc* between the two groups (Figs. 3C–3D, Fig. S3).

## RBM25 binds and regulates alternative splicing of genes associated with inflammation in H9c2 cells

We next conducted iRIP-seq to explore RBM25 binding capability and identify interacting transcripts. After confirming the success of immunoprecipitation by observing the difference in RBM25 expression levels between the two groups (Fig. 4A), we analyzed the uniquely mapped anti-RBM25 binding peaks in the reference rat genome. As depicted in Figs. 4B–4C, the reads of immunoprecipitated (IP) sample reads were predominantly enriched in the coding sequence and intrinsic regions, implying a high potential for pre-RNA binding. We next used hypergeometric optimization of motif enrichment (HOMER) to call RBM25-bound peaks and identify the top five binding motifs. The results revealed that the GA-rich motif was the most enriched in ABLIRC peaks, as displayed in Fig. 4D. Between the two experiments, we identified 561 overlapping peaks (Fig. 4E) and the top ten KEGG and GO biological processes are listed in Figs. 4F and 4G, respectively.

Considering the role of RBM25 in both transcription and post-transcription, we performed an integrated analysis between the iRIP-seq data and the RASGs. As a result, we identified 29 overlapped genes, as shown in Fig. 5A. The key functions of these overlapping genes, based on KEGG analysis, are presented in Fig. 5B. The top 30 genes were ranked based on the number of binding peak tags and the number and ratio of RBM25-OE DEG reads. Among these, we selected *Brd8, Coro6,* and *Csf1* to conduct further validation using RIP PCR (Figs. 5C–5D, Fig. S4). As expected, the differences in expression levels between the two groups were significant.

## DISCUSSION

Our research represents one of the first efforts to examine RBM25-RNA interactions in H9c2 cells using RNA-seq and iRIP-seq technologies. Our findings revealed that RBM25 not only interacted with the inflammation-associated gene AS in H9c2 cells, but also affected genes primarily involved in inflammation and viral response.

RBM25 contains an N-terminal RNA-binding motif rich in arginine and glutamic acid residues and a C-terminal domain enriched in proline, tryptophan, and isoleucine. RBM25 is known to act as a pre-mRNA processing factor that interacts with diverse splicing components and assists with the assembly of spliceosome complexes (*Zhou et al., 2008*). Previous studies have established that RBM25 exacerbates cardiomyocyte apoptosis by regulating Bcl-x pre-mRNA, leading to the truncation of the cardiac voltage-gated Na+ channel (*Zhou et al., 2008*; *Gao et al., 2011*). Furthermore, the AKT signaling pathway, a pathway that has been widely implicated in HF, has also been revealed to be partially modulated by RBM25 (*Mohammad et al., 2016*). The established role of RBM25 in HF highlights the need for an investigation into the precise mechanisms through which RBM25

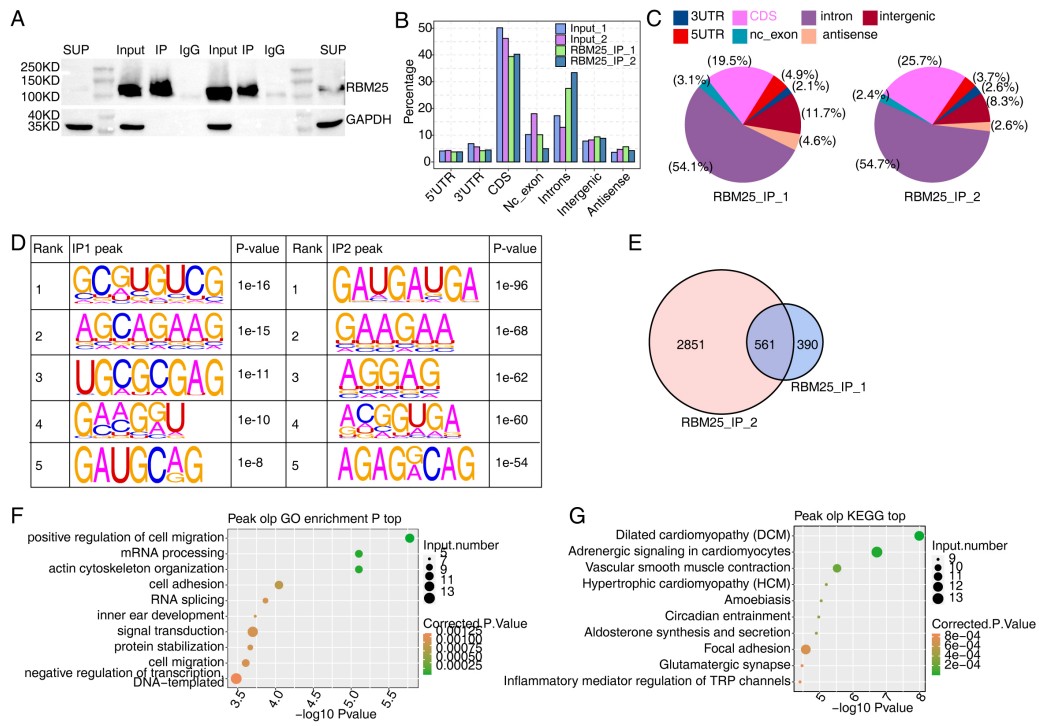

**Figure 4** **Characterization of the RBM25-RNA interaction profile by iRIP-seq analysis.** (A) Western blot experiment to verify IP efficiency. (B) Bar plot showing the distribution of the reads across reference genome. (C) Pie chart showing the distribution of the peaks across the reference genome. (D) Motif analysis showing the top 5 peaks preferred bound motifs of RBM25 by HOMER software. (E) Venn plot showing the overlapped peak of two IP samples. (F) Bubble diagram exhibiting the most enriched GO biological process of the overlapped peak genes. (G) Bubble diagram exhibiting the most enriched KEGG pathways of the overlapped peak genes.

impacts HF at the transcriptional or post-transcriptional stages of RNA processing. To address this, we sought to probe the function of RBM25 in H9c2 cells by analyzing RNA-seq and iRIP-seq data.

Our RNA-seq analysis suggested that the overexpression of RBM25 regulated inflammation- and viral response-related genes in H9c2 cells. Specifically, among all DEGs, *Lnc215, Irf7, Cxcl10, Oas1b, Irgm2, Mx2, Ifi47, Ifit1, Oas1a, Cd28,* and *Dhx58* had the highest expression level. Validation experiments yielded 100% accordance rate. In line with previous studies implicating *Irf7, Cxcl10, Ifit1/Ifit3,* and *Cd28* in the inflammatory and immune responses in HF and pathological cardiac hypertrophy (*Jiang et al., 2014*; *Altara et al., 2016a*; *Altara et al., 2016b*; *Wang et al., 2016*; *Becher et al., 2018*; *Cao et al., 2018*), we found these genes to be upregulated following RBM25 overexpression in our investigation. These findings suggest that RBM25 plays a crucial role in the development and progression of HF by regulating immune- and inflammation-related factors, thereby driving subsequent immune and inflammatory reactions. However, the exact signaling pathway through which RBM25 triggers inflammation in HF remains to be elucidated. Interestingly, we also found that several long non-coding RNAs (lncRNAs) with yet

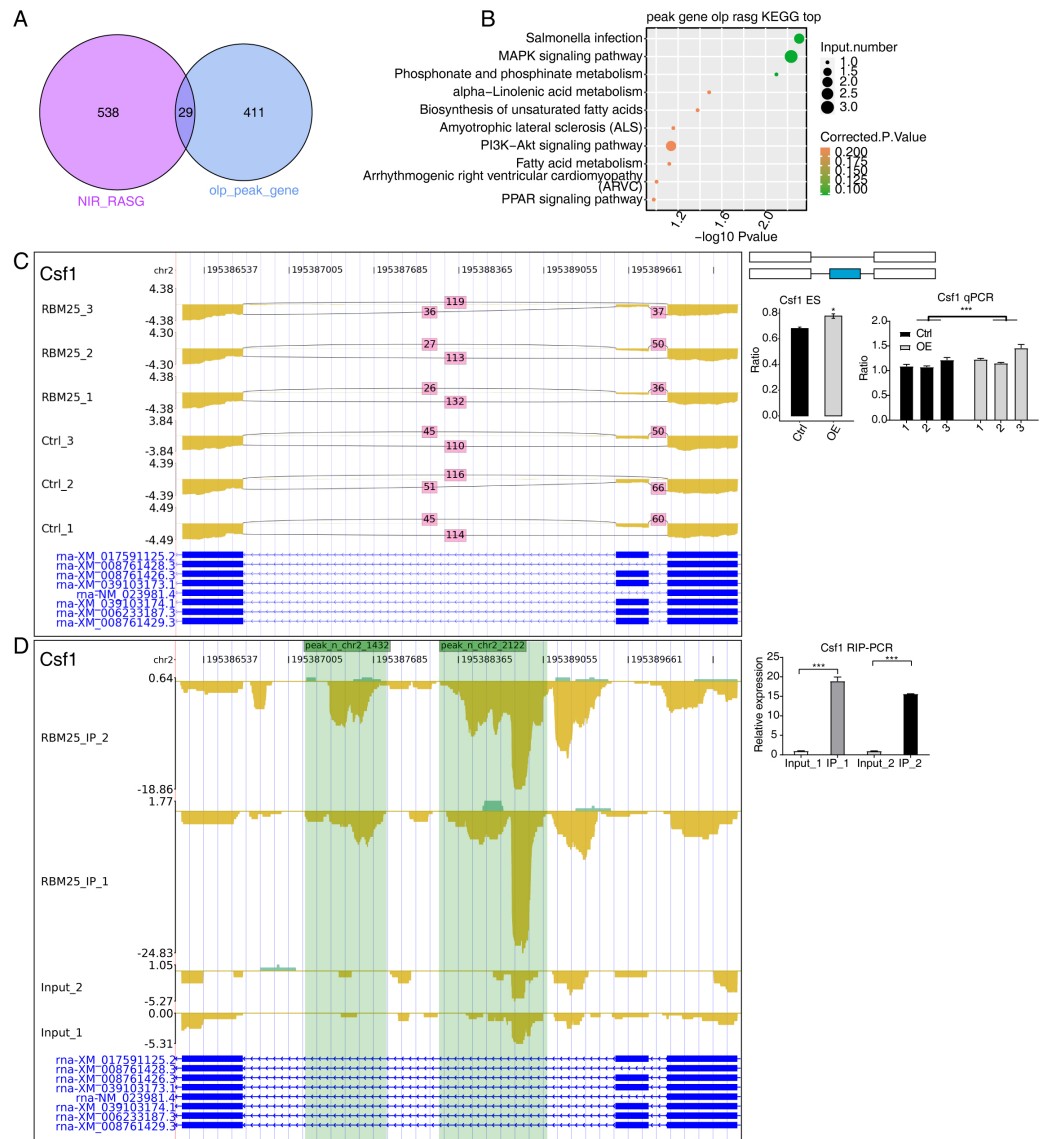

**Figure 5** **RBM25 binds to mRNA and regulates its alternative splicing.** (A) Venn diagram showing the overlapped genes of RBM25-bound genes and RASGs. (B) Bubble diagram exhibiting the most enriched KEGG results of the overlapped genes in (A). (C–D) IGV-sashimi plot showing the peaks reads and RBM25-regulated alternative splicing events and binding sites across mRNA of *Csf1*, the green panels represent the position of peaks. Reads distribution is plotted in the up panel, and each gene's transcripts are shown below. The schematic diagrams depict the structures of ASEs on the right panel. QPCR validation of ASEs and RIP-PCR validation of peaks are shown at the bottom of the right panel. Error bars represent mean ± SEM. *P value < 0.05, **P value < 0.01, ***P value < 0.001.

uncharacterized functions, including *Lnc215, Loc120097815,* and *Loc120099122,* were strongly associated with RBM25 overexpression in H9c2 cells. These findings hint at novel mechanisms in HF necessitating further exploratory studies.

AS, which plays a crucial role in post-transcription regulation, conspicuously influences gene expression patterns. mRNAs, being alternatively spliced, can translate into numerous

protein variants differing in structure and function. However, aberrant AS can lead to cellular dysfunction and disease, including cancer, neurodegenerative disorders, and autoimmune diseases (*Novak et al., 2009*; *Nik & Bowman, 2019*; *Bonnal, López-Oreja & Valcárcel, 2020*).

According to a groundbreaking study showing that RBM25 negatively impacts HF by promoting alternative splicing of SCNA5 (*Gao et al., 2011*), we focused our initial investigation on changes in ASEs after RBM25 overexpression. Analysis of splice junctions identified 147,716 junctions, which have the potential to generate new transcripts and influence protein expression. Nonetheless, the biological implications of these novel sequences in the context of RBM25 upregulation warrant further investigation. As anticipated, AS patterns varied in the RBM25-OE group compared to the control group; this indicated that RBM25 may influence HF by modulating AS in H9c2 cells. Particularly, the RASGs were found to predominantly participate in the cellular apoptosis pathway, supporting previous findings that RBM25 knockdown inhibited apoptosis, while upregulation of RBM25 exacerbated apoptosis (*Zhou et al., 2008*; *Ge et al., 2019*). RBM25, known for its role as an RNA-binding apoptosis regulator, partially modulates Bcl-x through dose-dependent activation of its 5′ splice site (5′ss) selection (*Zhou et al., 2008*). Bcl-x, a member of the Bcl-2 family, plays a fundamental role in governing apoptosis and guaranteeing its precision (*Li et al., 2020*). In cardiomyocytes, stimuli such as myocardial ischemia or pressure overload may trigger the apoptotic process, culminating in congestive HF (*Williams, 1999*). The mitochondria-dependent apoptotic pathway, characterized by a reduction in Bcl-2 protein and an elevation in Bax protein and its downstream gene *Casp3*, is currently a key research focus in HF (*Danial & Korsmeyer, 2004*). AS is considered indispensable for apoptosis modulation because key factors that undergo mRNA splicing during apoptosis can give rise to isoforms with enhanced functionality (*Schwerk & Schulze-Osthoff, 2005*). Moreover, there is evidence that RBM25 selectively interacts with *LUC7L3* at the CGGGCA exonic splicing enhancer cis-element binding site, thereby activating pro-apoptotic Bcl-x(S) 5′ splicing in HeLa cells (*Zhou et al., 2008*). As such, it is plausible that RBM25 could contribute to apoptosis-mediated HF in H9c2 cells through alternative splicing of apoptosis-related genes, such as *LUC7L3*.

Our RNA-seq data revealed significant differences in the aforementioned ASEs and their ratios between the RBM25-OE group and control group, highlighting the direct role of RBM25 in regulating AS within H9c2 cells. Among these genes, only a few pertained to HF, including ubiquitin-protein ligase e3a (*Ube3a*), *Il17rc*, and *Slc38a9*. Previous studies have acknowledged the strong correlation between *Ube3a*, a biomarker for myocardial hypertrophy, and apoptosis in H9c2 cells (*Song et al., 2015*; *Cheng et al., 2019*). *Il17rc*, a pro-inflammatory receptor, may escalate inflammation responses by elevating levels of IL-17, which contributes to left ventricular remodeling by inducing fibrosis, collagen production, and apoptosis. The MAPK signaling pathway, which is enriched in our RBM25-overexpressed H9c2 cells according to KEGG analysis, is implicated in these interactions (*Chang et al., 2018*). This finding corroborates our DEG results suggesting that inflammation and immune activation contribute to the onset and progression of HF. Additionally, the central amino acid transporter *Slc38a9*, another gene associated with

HF, stimulates mTORC1, provoking pathological hypertrophy and the accumulation of misfiled proteins in hearts under pressure overload, ischemic injury, and aging (*Wyant et al., 2017*; *Sciarretta et al., 2018*). Furthermore, the *Slc38a9*-dependent amino acid efflux sensed by mTORC1 can initiate a negative feedback loop of autophagy, a crucial process in degrading misfolded proteins and damaged organelles to maintain normal cardiac function. Nevertheless, this process tends to be repressed during HF (*Gatica et al., 2022*).

We further investigated the interactions between RBM25 and RNA in cardiomyocytes using iRIP-seq. The majority of sequencing reads of RBM25-associated RNAs were located within the intron region, with minimal enrichment in the initiation sites of transcription or translation. Our findings hint that RBM25 primarily operates *via* pre-mRNA binding, which further signifies its direct involvement in AS rather than its ability to influence the co-transcription or translation process. Through analyzing the motif of RBM25-specific binding peaks using HOMER, we found a consistent GA-rich motif across two repeated samples. Our results vary from previous research, which indicated that RBM25 interacted with the exonic splicing enhancer CGGGCA to facilitate activation of the proapoptotic Bcl-x(S) 5′ss (*Zhou et al., 2008*). We failed to detect the 5′-CGGGCA-3′ exon motif because the intron region covered more than 50% of the binding sites investigated in our study. RBM25 has distinct functions in HeLa and H9c2 cells, with the former predominantly impacting apoptosis. While the current study focuses on inflammation, future research should continue to explore additional functions of RBM25. Following functional analysis, we observed significant enrichment for positive regulation of cell migration, mRNA processing, actin cytoskeleton organization, cell adhesion, and RNA splicing. Therefore, we propose that AS regulation of RBM25-bound RNA, which is primarily enriched in cell migration and adhesion, RNA processing, and splicing, could underpin the unfavorable role of RBM25 in HF.

After mapping RBM25-regulated ASEs and binding peaks, we conducted an overlap analysis between DEGs from RNA-seq and repeated binding peaks from iRIP-seq. However, only *LOC12009412*, a gene with an as yet unspecific biological function, appeared consistently in both databases, although its role in the process of RBM25-mediated effects is yet to be clarified. Similarly, overlap analysis was performed between RBM25-regulated ASEs and iRIP-seq, which revealed 29 AS-related genes capable of binding to RBM25. Among these genes, *Coro6, Csf1*, ADAM metallopeptidase with thrombospondin type 1 motif 10, and *Brd8* have already been confirmed to participate in the pathophysiology of HF. Consequently, we selectively validated AS regulation on *Slc38a9, Il17rc, Csf1, Coro6*, and *Brd8* based on our findings. Except for *Il17rc*, which showed no significant difference in terms of expression value, all target genes conformed to the expected trends, thereby achieving an accuracy rate of approximately 80%. Subsequently, we selected three genes, *Coro6, Csf1*, and *Brd8*, for validation through both RNA-seq and iRIP-seq data using RT-qPCR. All three genes were validated with 100% accuracy. Existing literature posits *Coro6* as one of the cardiac target genes modulated by RBM24, a gene controlling most splicing events linked to cardiac development and disease (*Van den Hoogenhof et al., 2018*). *Coro6*, considered one of the most robust heart-specific markers indicative of hypermethylation, implies a predisposition to potential cardiac damage (*Ren et al., 2022*). The successive

verification of *Coro6* in our study provides further evidence to investigate the probable crucial role of *Coro6* in cardiomyocyte injury. Another gene, *Csf-1*, is a proinflammatory cytokine linked to multiple inflammation-related diseases. The production of *Csf-1* can trigger the proliferation of tissue-resident macrophages and reduce apoptosis (*Shaposhnik, Wang & Lusis, 2010*). Indeed, a previous study demonstrated that decreased *Csf-1* levels significantly mitigate the inflammatory response, leading to a reduction in cardiac fibrosis and enhanced ventricular function in myocarditis mouse models (*Meyer et al., 2018*). A similar finding has been reported in experimental autoimmune myocarditis, in which *Csf-1* was found to inhibit the accumulation of fibroblasts (*Blyszczuk et al., 2013*). The identification of *Csf-1* and its consequential regulation of pre-mRNA AS expands our understanding of HF. Regarding *Brd8*, as expected, the *Brd8*-IP group was significantly enriched at the primer-binding site. However, in contrast to our original hypothesis, the *Brd8* gene exhibited a statistically significant decrease in the AS ratio. The unexpected result for the *Brd8* gene requires research to understand whether it follows a more complex molecular interaction pattern or has previously unknown biological activity.

Several limitations in this study should be noted. First, our work used an overexpression design, which may understate the effects of RBM25 transfection in H9c2 cells due to the potentially toxic impacts of the gene transfection on cardiomyocytes (*Pugach et al., 2015*). Additionally, we failed to verify the localization of RBM25 in an overexpressed setting, which in theory should be in the nucleus. Furthermore, our study is largely based on bioinformatic analysis and lacks validation from animal experiments. Additional studies in animal models and humans could provide a more comprehensive understanding of the role of RBM25 in HF.

## CONCLUSIONS

In this preliminary study, we investigated the RNA-binding motif of RMB25 in H9c2 cells, as well as whether it regulates AS. Our results suggest that in H9c2 cells, RBM25 binds to and modulates the AS of genes that are crucial for inflammation, such as *Slc38a9, Csf1*, and *Coro6*. This investigation has set the stage for in-depth functional analyses and further characterization of the role of these genes in H9c2 cells.

## ACKNOWLEDGEMENTS

We thank ABLife for performing partial statistical analyses. We thank International Science Editing for editing this manuscript.

### Funding

This work was supported by the National Natural Science Foundation of China under grant [82360082] and [81760074]; the Special Foundation Projects of Joint Applied Basic Research of Yunnan Provincial Department of Science and Technology with Kunming Medical University under grant [202201AY070001-064] and [2017FE468(-043)]; the

Yunnan Health Training Project of High-Level Talents under grant [D-2018020]; the Clinical Medical Center for Cardiovascular and Cerebrovascular Disease of Yunnan Province under grant [ZX2019-03-01]; the Foundation Projects of young and middle-aged academic and technical leaders reserve talent of Yunnan province under grant [202305AC160048]. The funders had no role in study design, data collection and analysis, decision to publish, or preparation of the manuscript.

## Grant Disclosures

The following grant information was disclosed by the authors:
National Natural Science Foundation of China: 82360082, 81760074.
Special Foundation Projects of Joint Applied Basic Research of Yunnan Provincial Department of Science and Technology with Kunming Medical University: 202201AY070001-064, 2017FE468(-043).
Yunnan Health Training Project of High-Level Talents: D-2018020.
Clinical Medical Center for Cardiovascular and Cerebrovascular Disease of Yunnan Province: ZX2019-03-01.
Foundation Projects of young and middle-aged academic and technical leaders reserve talent of Yunnan province: 202305AC160048.

## Competing Interests

The authors declare there are no competing interests.

## Author Contributions

- Xin Tian performed the experiments, analyzed the data, prepared figures and/or tables, authored or reviewed drafts of the article, and approved the final draft.
- Guangli Zhou performed the experiments, prepared figures and/or tables, and approved the final draft.
- Hao Li performed the experiments, prepared figures and/or tables, and approved the final draft.
- Xueting Zhang performed the experiments, prepared figures and/or tables, and approved the final draft.
- Lingmin Zhao performed the experiments, prepared figures and/or tables, and approved the final draft.
- Keyi Zhang performed the experiments, prepared figures and/or tables, and approved the final draft.
- Luqiao Wang performed the experiments, prepared figures and/or tables, and approved the final draft.
- Mingwei Liu performed the experiments, prepared figures and/or tables, and approved the final draft.
- Chen Liu conceived and designed the experiments, prepared figures and/or tables, authored or reviewed drafts of the article, and approved the final draft.
- Ping Yang conceived and designed the experiments, prepared figures and/or tables, authored or reviewed drafts of the article, and approved the final draft.

## Ethics

The following information was supplied relating to ethical approvals (*i.e.*, approving body and any reference numbers):

This project is approved by the Institutional Ethics Committee at the First Affiliated Hospital of Kunming Medical University (kmmu20211238, Yunnan, China).

## Data Availability

The RNA-seq and iRIP-seq original data is available at NCBI GEO: GSE209940 and GSE209941.

## Supplemental Information

Supplemental information for this article can be found online at http://dx.doi.org/10.7717/peerj.16312#supplemental-information.

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
