# Peer review of "RBM25 binds to and regulates alternative splicing levels of *Slc38a9, Csf1,* and *Coro6* to affect immune and inflammatory processes in H9c2 cells"

_PeerJ, doi:10.7717/peerj.16312_

## Round 0.1 · original submission · Major Revisions

Please address the concerns of all reviewers and revise the manuscript accordingly.

·

Basic reporting

This article is riddled with various formatting and grammatical errors. The writing style exhibits a lack of professionalism and consistency. Additionally, the figures and legends within the article do not accurately correspond to the content. It is highly recommended to seek the assistance of a professional editing service to ensure the manuscript is free from errors and presented in the best possible manner.

Experimental design

This study is well-designed, and the initial scope and objectives of the research have been clearly outlined.

Validity of the findings

The findings presented in this study have effectively addressed the research question.

Additional comments

Major concerns:
Since this study is based on the overexpression of pCDNA3.1-3xflag-RBM25, the transfection efficiency directly impacts the reliability of all experiments. Therefore, it is advisable to establish a stable cell line using the lentivirus system to conduct this study.

Minor concerns should be addressed:
1. Since there are six figures at the end of the manuscript, in the Figure Legend section, Figures should be labeled as Figure 1 to Figure 6. Therefore, Figure 2-2 should be corrected to Figure 3. The label 'Figure 2-2' is not professional. Accordingly, all Figure labels in the main text should be changed accordingly.
2. In line 250, change '(figure 2C)' to '(Figure 2C)' to maintain consistent formatting throughout the paper.
3. In line 265, it is mentioned 'Figure 2F,' but it should be corrected to 'Figure 3.' There is no Figure 2F in manuscript.
4. In the Results section, either delete lines 200 to 227 or move them to the Discussion section.
5. In the legend of Figure 1, statistical methods should be included.
6. In Figure 3 and Figure 4, the positions of Ctrl and OE in the chart should be placed below the chart and should not overlap with the chart itself. Additionally, in the legend of Figure 3 and 4, the mean of **** should be included in the legend section.
7. In Figure 4 and 6, the agarose gel electrophoresis of the PCR should be included to demonstrate the changes in isoforms.

Reviewer 2 ·

Basic reporting

No comment

Experimental design

No comment

Validity of the findings

No comment

Additional comments

In the submitted manuscript, the authors described the regulation of inflammatory response genes' alternative splicing in RNA binding motif 25 overexpressed H9c2 cells. I agree with the authors that this work will pave the way for a deeper functional analysis of the mentioned genes' role in HF inflammatory response. The analysis shows that the authors possess extensive expertise regarding their experimental studies and effectively showcased it in a structured manner.
However, I provided some corrections and raised a few inquiries in order to enhance the quality of the manuscript.

Annotated reviews are not available for download in order to protect the identity of reviewers who chose to remain anonymous.

Reviewer 3 ·

Basic reporting

The manuscript by Tian et al., explores the role of RBM25 in the regulation of alternative splicing and gene expression in cardiomyocyte derived cell line H9c2. The study uses an exogenously overexpressed RBM25 system in H9c2 cells followed by steady-state RNA-seq and RBM25 RIP-seq. The authors correlate their findings with pathophysiology of heart failure.

The manuscript “RBM25 binds to and regulates alternative splicing levels of Slc38a9, Il17rc, Csf1, and Coro6 to affect the immune and inflammatory process in H9c2 cells” is a preliminary study with both conceptual, technical, and experimental flaws. Above all, the manuscript is extremely vaguely written and needs extensive revisions before acceptance anywhere.

While there is no mention of shRNAs either in the method or result sections, the discussion lines 355-356 describes the use of RBM25 specific shRNAs, if this is the case the interpretation of data presented in the entire manuscript would be different unless authors mean that they somehow depleted endogenous rbm25 while overexpressing exogenous RBM25.

The existing algorithms of alternative splicing analysis are not 100% reliable and need extensive analysis by multiple algorithms like rMATS, Majiq etc. while the current manuscript solely relies on the HISAT2.

Line 275-276: 147716 novel splice junctions were identified, what these junctions are and how these junctions impact the proteome of cells is not explained anywhere.

While studying the impact of a protein on its regulatory functions, the knockdown or knockout approaches are primarily used instead of the overexpression of proteins.

Extensive spelling and grammatical mistakes, use of jargon words etc. make it extremely difficult to follow the story and eventually diminishes the actual impact of the work. Particularly methods are extremely poorly written.

Few examples include:
Line 42: Use of both present and past tenses in the same sentence (upregulates, modulated).
Line 37: Excessively expressed. Should be exogenously expressed or overexpressed.
Line 78: scanty
Line 84: sort of proteins
The SCNA5 is written as SCNA5 in line 98 and s SCN5A in line 100.
Line 116: perchaed
Line 136: what are the prior instructions, does this mean prior study?
Line 179: How this sentence couples the previous study to the current.
Lines 192-195: Use of present tense.
Line 200: Primer binding site was employed as a control gene to verify specific genes detected in iRIP. What does this mean?
Use of word obviously.
Line 245: The fold change cutoff value in line 245 (> 2.0 or < 0.5) is different than figure legend 2B (> 1.5 or < 0.67).

There is no figure 2F, figure labelling beyond 2F is not correct.

These are only a few examples of how the manuscript needs careful writing and language improvements.

Unfortunately, the manuscript is week in its current form which could be improved in future by extensive reading/writing efforts and careful interpretation of the data.

Experimental design

The existing algorithms of alternative splicing analysis are not 100% reliable and need extensive analysis by multiple algorithms like rMATS, Majiq etc. while the current manuscript solely relies on the HISAT2.

While studying the impact of a protein on its regulatory functions, the knockdown or knockout approaches are primarily used instead of the overexpression of proteins.

Methods are vaguely written.

Validity of the findings

While the q-PCR based validation is fine, the authors should show the differential spliced isoforms on agarose gels.

Reviewer 4 ·

Basic reporting

-The manuscript has merit and is backed by supporting data and literature references. However, there is a need to QC the manuscript thoroughly for typos, grammar and sentence structure which will significantly improve the quality of the manuscript and make the findings more easy to comprehend for the readers (see attachment for suggestions).
-Literature references are largely sufficient and figures/ tables and supplementary data provided are easy to follow; Some improvement can be made to figure legends to clearly define what is presented in the figure.

Experimental design

-The manuscript presented is original with a clearly defined problem statement and a gap in the field that the research is trying to address.
-While data presented is largely robust, some additional analyses are recommended with already available datasets to fully support the claims made in the paper (see attachment for recommendations).
-Materials and Methods are well described but need some language refinement (see comment under basic reporting)

Validity of the findings

Data presented to support the findings are fairly sound. I would recommend the authors to tone down the final conclusions of the paper in the abstract and limit claims to actual findings that are supported by experimental results. Moreover, I recommend the authors to add some additional information on certain experimental controls (see attachment for details), to fully support the conclusions.

Additional comments

No additional comments

Annotated reviews are not available for download in order to protect the identity of reviewers who chose to remain anonymous.

---

## Round 0.2 · Minor Revisions

Please address the remaining issues pointed out by the reviewer and amend the manuscript accordingly.

·

Basic reporting

no comment

Experimental design

no comment

Validity of the findings

no comment

Additional comments

I acknowledge and appreciate the authors' sincere efforts to bring to this manuscript.The authors have done a very good job of addressing the previous concerns.

Reviewer 2 ·

Basic reporting

No comment

Experimental design

A new error is introduced while fixing an older mistake.
Please check line 191, and 193 "(50 nM Tris 8.0, 10 mM EDTA, and 1% SDS, pH = 7.4)". Explain or remove 8.0 after Tris.

Validity of the findings

No comment

---

## Round 0.3 · Minor Revisions

Thank you for addressing remaining issues. While checking your resubmitted manuscript, the Section Editor made the following observations:

- the language still needs a lot of work (sentence starting with "remains"; purchase as "perchaed" or provided as "providede")

- the methods are very poorly described

- just one algorithmic analysis is insufficient. Where are the primers? And a t-test to analyse these data?

Please address these concerns and revise the manuscript accordingly.

---

## Round 0.4 · accepted · Accept

All remaining concerns are addressed and the manuscript is revised accordingly.